# Moderate Reduction in Nitrogen Fertilizer Results in Improved Rice Quality by Affecting Starch Properties without Causing Yield Loss

**DOI:** 10.3390/foods12132601

**Published:** 2023-07-05

**Authors:** Yimeng Li, Chao Liang, Junfeng Liu, Chanchan Zhou, Zhouzhou Wu, Shimeng Guo, Jiaxin Liu, Na A, Shu Wang, Guang Xin, Robert J. Henry

**Affiliations:** 1College of Agronomy, Shenyang Agricultural University, Shenyang 110866, China; liym2019@stu.syau.edu.cn (Y.L.); liangchao@stu.syau.edu.cn (C.L.); junfengliu58@163.com (J.L.); zhouchan@syau.edu.cn (C.Z.); 2022200082@stu.syau.edu.cn (Z.W.); 2022200067@stu.syau.edu.cn (S.G.); 2020220276@stu.syau.edu.cn (J.L.); 2021220269@stu.syau.edu.cn (N.A.); 2Queensland Alliance for Agriculture and Food Innovation, The University of Queensland, Brisbane 4067, Australia; robert.henry@uq.edu.au; 3College of Food Science and Engineering, Shenyang Agricultural University, Shenyang 110866, China; xguang212@163.com

**Keywords:** fine amylopectin structure, texture, structure and properties, rice starch, nitrogen application

## Abstract

The quality and starch properties of rice are significantly affected by nitrogen. The effect of the nitrogen application rate (0, 180, and 230 kg ha^−1^) on the texture of cooked rice and the hierarchical structure and physicochemical properties of starch was investigated over two years using two japonica cultivars, Bengal and Shendao505. Nitrogen application contributed to the hardness and stickiness of cooked rice, reducing the texture quality. The amylose content and pasting properties decreased significantly, while the relative crystallinity increased with the increasing nitrogen rates, and the starch granules became smaller with an increase in uneven and pitted surfaces. The proportion of short-chain amylopectin rose, and long-chain amylopectin declined, which increased the external short-range order by 1045/1022 cm^−1^. These changes in hierarchical structure and grain size, regulated by nitrogen rates, synergistically increased the setback viscosity, gelatinization enthalpy and temperature and reduced the overall viscosity and breakdown viscosity, indicating that gelatinization and pasting properties were the result of the joint action of several factors. All results showed that increasing nitrogen altered the structure and properties of starch, eventually resulting in a deterioration in eating quality and starch functional properties. A moderate reduction in nitrogen application could improve the texture and starch quality of rice while not impacting on the grain yield.

## 1. Introduction

Rice (*Oryza sativa* L.) is the only staple food that is consumed mostly in the form of whole grains after cooking [1]. Rice quality is a very complex trait including four aspects; namely, milling quality, appearance quality, cooking and eating quality, and nutritional quality [2,3]. Among these, cooking and eating quality has become the principal characteristic of concern for breeders and consumers [4]. Genetic, environmental, and agronomical factors have been reported to be mainly responsible for variation in rice composition and eating quality [5,6].

Nitrogen fertilizer application is a typical agronomic strategy used to increase yield [7] and to regulate rice quality [8] because both the yield and quality are directly determined by grain filling. Previous research showed that nitrogen application affected the grain filling of rice by prolonging the duration of the grain-filling period, regulating the starch biosynthesis process [9], resulting in variation in the hierarchical structure and physicochemical properties of starch [10,11], and, ultimately, influencing the grain plumpness [12].

Nitrogen fertilizer application improves the grain yields of rice but reduces the eating quality of rice. This may be caused by an increased number of spikelets per unit area, resulting in less photosynthate for each grain, and a higher percentage of inferior grains that are not totally filled. The starch quality and eating quality will certainly decline as a result of the undeveloped inferior grains [12]. Increases in nitrogen fertilizer application cannot be used to linearly increase rice yield in production practices [13]. Furthermore, high nitrogen levels are not only inconsistent with sustainable agricultural production but also increase risks, such as pests and diseases, lodging, and soil organic matter loss [14]. However, it should be possible to achieve a stable yield and high quality by appropriately reducing the amount of reduced N fertilizer.

Starch, which is approximately 90% of the endosperm weight, plays a crucial role in the eating quality and texture characteristics of cooked rice [1,6,11]. Starch consists of two main molecular components, amylose and amylopectin. The content of amylose [15] and the distribution of amylopectin [16,17] have been proven to be closely related to the quality and texture properties of cooked rice. The differences in starch include variation in the morphology of granules (shape, size, and presence of compound granules), surface attributes (surface pores), the existence of non-starch components (proteins, lipids, and cell wall remnants), and molecular composition and conformation (size and quantity of amylose and amylopectin, type and degree of crystallinity), as described elsewhere [18,19,20,21,22], and are responsible for the quality and texture properties of cooked rice.

Extensive reports have been published regarding the effect of nitrogen fertilizer on the properties of starch derived from diverse biological sources, including studies of common buckwheat [23,24], wheat [25], maize [26], and rice [8,27]. In a soft super rice production experiment, moderate nitrogen fertilizer reduction was found to improve the quality while still maintaining the yield [13]. However, there is limited information on how nitrogen affects eating quality and the characteristics of *japonica* rice starch.

We hypothesized that the nitrogen application rate may change the starch properties and cooked rice texture properties by altering the starch granules’ size and fine structure of amylopectin, as well as the ratio of amylose and amylopectin. A controlled field nitrogen experiment was conducted in Panjin, Liaoning, China. The objectives were to evaluate the effect of a reduction in the nitrogen application rate on (i) the composition of the grain and textural properties, (ii) the fine structure of amylopectin, and (iii) the thermal and pasting properties of rice starch. The results will enhance the understanding of rice starch, and how to improve the starch quality and rice quality to conform to the concept of sustainable agriculture by reducing nitrogen application.

## 2. Materials and Methods

### 2.1. Rice Plants and Experimental Design

Three replicates of two *japonica* rice cultivars, Bengal and Shendao 505 (SD505), were cultivated in the field at Xi’an farm in Panjin, Liaoning province, China (122°31′ E, 41°22′ N) during the rice cultivation season in 2020 and 2021. The Liaohe River Delta plain is a major rice growing region where the rice planted takes up 15% of the land in Liaoning. With the advantages of river irrigation and climate, rice produced in Panjin has a better quality and palatability than rice from other places. The soil type is littoral saline paddy soil with 26.4 g kg^−1^ of organic matter, 82.57 mg kg^−1^ of available nitrogen, 13.28 mg kg^−1^ of available phosphorus, 215.83 mg kg^−1^ of available potassium and a pH of 8.10. The daily temperature and precipitation from transplanting to harvesting are shown in Appendix A for 2020 and 2021. 

The experiment setup employed a split plot design, with nitrogen application level as the main plot and cultivar as the split plot. Seeds were sown on 18 April 2020 and 20 April 2021 and sequentially transplanted on 25 May 2020 and 26 May 2021, respectively. Three seedlings per hill with a spacing of 30 cm × 18 cm were transplanted. The plot dimensions were 17 m × 3 m. A total nitrogen amount of 0, 180, and 230 kg ha^−1^ was applied as basal, striking root, and tillering fertilizers with proportions of 65%, 15%, and 20%, respectively. Phosphorus (90 kg ha^−1^ as superphosphate) and 50% potassium (110 kg ha^−1^ as K_2_SO_4_) were applied and incorporated as a basal fertilizer before transplanting. The other potassium was applied during the tillering period. Other practices were carried out according to local recommendations. The total nitrogen application of 230 kg ha^−1^ was the application practice of local farmers.

### 2.2. Starch Isolation

The rice grains were processed using a rice huller, followed by three rounds of grinding into fine flour using a DFT-small-sized flour milling machine (Xuanfeng, Shenyang, China). The rice flour was sieved (100-mesh) and stored for further use. 

Starch isolation was conducted based on the method described by Zhu et al. (2016) with minor modifications. To eliminate protein in the rice flour, 5 g of the sample was steeped in a pH 9–11 aqueous solution (40 mL) of sodium hydroxide containing 0.05% of alkaline protease (200 u g^−1^ enzyme activity Macklin/Macklin Shanghai, Shanghai, China) at a temperature of 42 °C for 24 h. The starch slurry obtained was then sieved through a 200-mesh sieve, and the residue collected on the mesh was mixed with 30 mL of deionized water. After stirring for 2 min, further sieving was carried out to minimize the presence of residue. The filtrates were combined with the initial filtrate, discarding any residue remaining on the mesh. The combined starch slurry filtrates were centrifuged at 4000 g for 20 min. After discarding the supernatant, the faint yellow upper layer was removed, and the remaining precipitate was re-suspended in 20 mL of deionized water. Centrifugation at 4000 g for 20 min was repeated, followed by the removal of the supernatant. To ensure the thorough removal of impurities, the centrifugation process was repeated five times. Subsequently, the starch was dried at 35 °C under ambient pressure and filtered through a 200-mesh sieve. The flour and starch samples were placed into plastic zip-lock bags and stored at 4 °C until further use.

### 2.3. Grain Yield and Compositions

The grain yield was assessed by harvesting all plants within a 3 m^2^ site (except border plants) in each plot. The recorded yield values were adjusted to 14.5% moisture content uniformly before statistical analysis. 

The total starch content (TSC) was measured using a commercial kit (Megazyme International Ireland Ltd. Co., Wicklow, Ireland) and the AOAC Method 996.11 (AACC Method 76-13.01). The apparent amylose content (AAC) was measured using the iodine colorimetric method [28]. The AAC was determined by measuring the absorbance at 620 nm utilizing a standard curve established with corn amylopectin and potato amylose as a reference. The crude protein content (PC) was estimated from the nitrogen content of the rice flour using the Kjeldahl method employing a nitrogen analyzer (Tecator Kjeltec 8400, FOSS, Hoganas, Sweden). The nitrogen content was converted to PC by multiplying it by 5.95 (the protein conversion coefficient of rice).

### 2.4. Texture Profile Analysis of Cooked Rice (TPA)

The polished rice (5 g) was placed into an aluminum cylinder with 7.5 g of water, rinsed after 0.5 min, and then soaked for 30 min. The rice was cooked with an electric cooker by heating it to 100 °C for 30 min followed by 10 min of steaming. The rice was gently stirred to loosen it, and then it was transferred to a cooling device for 30 min with a filter paper cover and kept at room temperature for 90 min. 

A texture profile analyzer (CT3-10K Brookfield, San Jose, CA, USA) was employed to measure the texture attributes of the cooked rice using a two-cycle compression, force versus distance program [16], according to the method of previous research [29] with some modification. A complete kernel was selected from the middle of the aluminum can and arranged on the base plate. A 20 mm long cylindrical probe (TA4/1000) was positioned 5 mm above the base plate with a TA-RT-KIT fixture, 50% target deformation, and 0.5 mm/s testing speed at a 5 g trigger load. The test was repeated on 10 individual samples for increased reliability and accuracy. 

### 2.5. Observations and Measurements of Starch

#### 2.5.1. Starch Granule Morphology 

The morphology of starch granules was studied by scanning electron microscopy (SEM-S4800, Hitachi, Tokyo, Japan). Rice starch was affixed to specimen stubs and coated with a layer of gold. Microscopic images were captured at an accelerating voltage of 5.0 kV and a magnification of ×2000.

#### 2.5.2. Particle size Analysis

The granule size was established using a laser diffraction particle size analyzer (Master ZS90 2000, Malvern, UK) according to a method described previously [30]. The starch was placed in absolute ethyl alcohol and stirred at a speed of 2000 rpm. The instrument was capable of measuring particle sizes ranging from 0.1 to 2000 mm. The distribution of granule size was characterized in terms of volume, number, and surface distribution.

#### 2.5.3. X-ray Diffraction (XRD) Analysis 

XRD patterns were scanned using a D8 Advance X-ray diffractometer (Bruker-AXS, Karlsruhe, Germany). The starch samples were analyzed under the following conditions: a current of 40 mA, a voltage of 40 kV, and a diffraction angle (2θ) range of 3–40° with a scanning speed of 4° min^−1^. The degree of relative crystallinity was calculated by using MDI Jade 6.5 (Material Data, Inc., Livermore, CA, USA).

#### 2.5.4. Fourier Transform Infrared (FTIR) Analysis 

The starch was analyzed with a Fourier transform infrared spectrometer (Bruker VETEX80, Ettlingen, Germany) following the method of a previous study [31]. To investigate structural variations at a molecular level, the spectra were deconvolved (Lorentzian assumed line shape, k factor of 1.9, half-width 19 cm^−1^) by OMNIC software 8.2 (THERMO Nicolet, Waltham, MA, USA) over a range from 1200 to 800 cm^−1^. Intensity measurements at 1045, 1022, and 995 cm^−1^ were performed on the deconvoluted spectra.

#### 2.5.5. Amylopectin Chain-Length Distribution (CLD)

Acetate buffer (pH 4.4, 0.6 M) and isoamylase (Sigma-Aldrich, Shanghai, China 1400U, 10 mL) were used to debranch the starch (5 mg, 1% *w*/*v*) at 37 °C after it had been dissolved in a boiling water bath for 60 min. The branch chain-length distribution of amylopectin was assessed with an ICS-5000 high-performance anion-exchange chromatograph (ICS5000+, Thermo Fisher Scientific, San Jose, CA, USA) using a Dionex^TM^ CarboPac^TM^ PA100 anion-exchange column (4.0 × 250 mm; Dionex) according to the method described by Zhou et al. (2015).The experimental conditions included a flow rate of 0.4 mL min^−1^, an injection volume of 5 μL, and a solvent system consisting of 0.2 M NaOH and 0.2 M NaAc.

#### 2.5.6. Determination of Pasting and Thermal Properties 

The starch pasting properties were tested using a rapid viscosity analyzer (RVA) (Model 3D, Newport Scientific, Warriewood, Australia) according to the method of Lu and Lu [32]. Briefly, rice flour at a concentration of 12% (*w*/*w*) was mixed at a constant stir speed of 160 g using the standard profile. The temperature profile employed was as follows: the start temperature was 50 °C held for 1 min, followed by heating to 95 °C at 11.84 °C/min, held at 95 °C for 2.5 min, and then cooling at the same speed to 50 °C and held for 1 min.

The thermal properties were investigated by a differential scanning calorimetry (DSC) with a Model 200 F3 Maia (Netzsch, Selb, Germany). Using an empty aluminum pan as a reference, the ratio of the weight of starch to water was 1:3 for each sample, which was sealed in a hermetic aluminum pan and scanned from 30 to 95 °C at a heating rate of 10 °C/min after overnight equilibration at room temperature [33]. The thermal transitions of starch samples were defined as To (onset temperature), Tp (peak of gelatinization temperature), and Tc (conclusion temperature), and ΔH referred to the enthalpy of gelatinization.

#### 2.5.7. Swelling Power and Water Solubility

Swelling power and water solubility were measured according to the method of Zhu et al. [33]. The starch was weighed (W0) and mixed with water (2%, *w*/*v*) then heated in a water bath at 95 °C for 30 min. The sample was cooled, centrifuged at 8000 g for 10 min, the supernatant was discarded, the colloid was weighed (W1), and the sediments were dried to a consistent weight (W2). The swelling power = W1/W2 (g g^−1^) and the solubility (%) = (1 − W2/W0) × 100%.

### 2.6. Statistical Analysis

All the data reported are presented as the average of three replicate observations unless otherwise specified. Duncan’s new multiple range test was used to analyze the variance of the experimental group means (*p* < 0.05) using the data processing system DPS 9.50 (Ruifeng, Hangzhou, China). Graphs were produced using Origin 2021 (OriginLab, Northampton, MA, USA).

## 3. Results and Discussion

### 3.1. Effect of N Rates on Yield 

The yields of the two cultivars in the two years are shown in Figure 1A. The application of nitrogen significantly increased the yield of grain, consistent with previous studies [10,13,18]. The two-year average for the yield of Bengal was increased by 50 and 48%, and SD505 increased by 60 and 87% in the N1 and N2 treatments compared with the N0 treatment, respectively. There is significant difference in N0, N1, and N2, but the difference is non-significant for the N1 and N2 treatment in both years, indicating that a moderate reduction in nitrogen application did not have a significant effect on grain yield, which means that a moderate reduction in nitrogen application could maintain the yield at a stable level. The differences between the results for the two seasons may be due to factors such as the weather conditions. The grain yield increased with increasing N fertilizer application but the yield increments gradually decreased, and after reaching a certain N application level, the grain yield declined [34,35]. High nitrogen application means a high input, higher risk of pests and diseases, is environmentally unfriendly, and contributes to lodging and other crop failure risks.

### 3.2. Protein, Total Starch, and Apparent Amylose Content 

Nitrogen application significantly increased the protein content and decreased the AAC, as shown in Figure 1C,D. Applying N fertilizer prolonged the filling period and increased the filling rate. N also affected the enzymes related to starch and protein synthesis, consequently affecting the synthesis and accumulation of starch and protein [36]. Amylose content was influenced by the status of starch accumulation, and decreased with poor grain filling [37].

Nitrogen significantly increased the protein content of polished rice [11], increasing from 6.6 to 7.9% on average as the nitrogen applied increased from N0 to N2. This may be due to the activities of key enzymes in nitrogen metabolism involved in grain filling [38] or the availability of nitrogen and amino acids. The examination of the genotypic differences in response to nitrogen application showed that SD505 was more sensitive to nitrogen application than Bengal and had a higher protein content. Rice flour from SD505 showed a higher protein content, increasing by 5.7–11.7% and 20.9–26.0% in N1 and N2, in the two years, respectively, compared with that produced without nitrogen application, while a 2.9–3.5% and 15.1–19.0% increase was observed in Bengal. 

The trend of AAC was opposite to that of protein content. Both total starch content and AAC tended to decrease with increasing N application, with a slight average decrease of 1.3% in the total content and an average decrease of 4.7% in the AAC. The change in amylose content was partly attributed to a subtle decrease in the total starch content and mostly attributed to the ratio of amylose to amylopectin. This phenomenon is consistent with the results reported previously [39,40]. Consistent with previous reports [6,27], an increase in the nitrogen application rate decreased the amylose content gradually. These changes in amylose, even if seemingly minor may have a substantial effect on starch functionality, as a variation of 1% in the amylose content significantly alters the gelatinization and thus pasting properties [41]. The AAC in 2020 was lower than that in 2021 for both the cultivars, which could be attributed to variations in the environmental conditions (e.g., temperature and light) during grain development [42]. Previous research has established that environmental stresses, specifically temperature, during the grain-filling stage induce modifications in the starch composition of rice grains [43]. For example, the temperature post anthesis significantly affects amylose synthesis [44]. Therefore, the starch content and amylose content may vary between years and should be compared in the same season. 

In terms of the chemical composition of the grains, the application of N fertilizer led to an increase in protein content and a decrease in amylose content, which is consistent with the results of previous studies [30].

### 3.3. TPA of Cooked Rice

The TPA is a common instrument which is used to obtain the force-displacement curve by a double-compression test, providing a direct link between the mechanical properties of food and its texture profile [1,45]. The TPA parameters of cooked rice at different nitrogen application rates were significantly different, especially for hardness and stickiness (Figure 1). Hardness, cohesiveness gumminess, and chewiness progressively increased, while stickiness and adhesiveness decreased with an increase in the nitrogen application rate (Appendix A), consistent with the results of Singh et al. [27]. According to the hardness and stickiness data, no significant difference was found between the N1 and N0, but there was a significant difference between the N2 and N0, suggesting that rice texture can be improved with moderate nitrogen reduction. Hardness can be expressed in terms of the maximum force on the first compression and stickiness is the negative force of the first cycle. These two attributes of cooked rice were the two main texture attributes, which have been related to the amount and type of leached starch molecules [46]. During cooking, rice absorbs water and the starch granules swell and are ruptured, which results in starch leaching out and induces a decrease in rice hardness. The amylose and short-chain amylopectin leached during cooking are considered to contribute to the hardness and stickiness of cooked rice, respectively [1]. Cooked rice with a higher hardness has been recognized as being correlated to a higher AAC, which prevents swelling and helps maintain the integrity of swollen starch granules [47]. In contrast, in this study, N0 showed a lower hardness, which is a strong indication that hardness is not only related to AAC but is a complex trait affected by many factors. The significant differences in texture parameters among nitrogen application rates could be attributed to the amylose content, fine amylopectin structure, granule size, and the content of protein which adhered to the starch surface. The decrease in the average granule size of starch with an increase in the nitrogen application rate may be one of the reasons for the increase in hardness (see Section 3.4.2). Rice cultivars with larger granules have been reported to have a lower hardness [27]. 

### 3.4. Effect on the Structure of Starch

#### 3.4.1. Granule Morphology

The variability in the morphological attributes of starch granules is contingent upon both the cultivars and the environmental conditions [48]. In the present investigation, the morphological characteristics of the extracted starch were thoroughly assessed utilizing a SEM and the micrographs are shown in Figure 2. For all the treatments the starch granules were in irregular polyhedral shapes and packed tightly together. Nitrogen application affected the appearance and morphology of starch in both cultivars. The granules of starch without nitrogen application (N0) were more uniform with a relatively smooth surface. This is in line with previous reports [30,49]. Indentations and pores were observed on the surface of the granules, and these became uneven with increasing nitrogen application. More small granules adhered to the larger ones in the N1 and N2 treatments. The changes in the morphology of the granules were ascribed to the difference in the grain filling processes, consequently leading to changes in various starch properties.

#### 3.4.2. Granule Size Distribution

Starch granule size distribution is an important factor that influences rice eating and cooking quality [50,51]. The effect of the nitrogen application rate on the fine structure of amylopectin and amylose formation likely contributes to the diversity of starch granule size [52]. The effect of different nitrogen application rates on starch granule volume, number, and surface area distribution is shown in Figure 3. Generally, the number distribution of granule size displayed a unimodal trend [30] with a peak at approximately 0.7 μm. The volume and surface area distribution of starch granule size displayed a bimodal pattern [27,29,53,54] with peaks at approximately 1 and 7 μm, respectively. Following the size classification described previously [29], starch granule sizes were grouped into A-type granule (≤2.6 μm) and B-type granule (>2.6 μm) categories in Table 1. Compared with those in N0, the starch granules in N1 and N2 exhibited a smaller size, including the surface, number, volume-weighted mean diameter for d (0.1), d (0.5), and d (0.9) in Table 2. As exhibited in previous research, a significantly higher content of small granules was observed with an increasing nitrogen application rate, and the mean particle size fell. Previous literature has reported that nitrogen levels could affect granule distribution by affecting the grain filling process, prolonging the duration [3], and delaying the development of amyloplasts [55]. Waduge [56] suggested that large granules appeared first during the grain-filling period and were decomposed into medium-size starch granules and small starch granules later. This may explain the increasing percentage of A-type granules in N1 and N2.

#### 3.4.3. X-ray Diffraction (XRD) and Relative Crystallinity (RC)

The XRD patterns and relative crystallinity (RC) at different nitrogen application levels were significantly different, as shown in Figure 4. The starches of all samples showed an A-type crystallinity displaying strong reflection peaks at 15.1° and 23° (2θ), and unresolved doublet peaks at 17° and 18°, which confirmed results with natural cereal starches [57]. Starches with nitrogen treatment showed a higher RC value than those without nitrogen and the value increased with the nitrogen rate. The results suggest that the application of nitrogen did not change the crystal type of rice starch, but enhanced the relative crystallinity of starch gradually, supporting the results of Zhu et al. [6]. The intensity increase may be due to the reduced amylose content, the ratio of short- to long-chain amylopectin [58], and the smaller granule size [59], which led to the compact arrangement of lattices and the formation of double-helix structures. These findings were consistent with the observed changes in starch helical structures and crystallinity regions (Section 3.4.4). The results in this study were similar to Cheetham’s finding [60], that relative crystallinity was positively correlated with a small granule proportion and the ratio of short- to long-chain amylopectin, while negatively correlated with AAC and average chain length, causing crystalline lamellae to be formed by amylopectin. 

#### 3.4.4. Fourier Transform Infrared Spectrum (FTIR)

FTIR was used to evaluate the short-range ordered structure in the external region of the starch granules [61]. The absorbance at 1045, 1022, and 995 cm^−1^ from the deconvoluted FTIR spectra were three typical characteristic bands of native starch, which were sensitive to the crystalline regions, amorphous regions, and the bonding in the helical structures in the carbohydrate, respectively [18,25] (see Figure 5). Hence, the ratios of 1045/1022 cm^−1^ and 995/1022 cm^−1^ can serve as quantitative indices to determine the degree of short-range molecular order in the external region of the granules [62,63].

The higher ratio of 1045/1022 cm^−1^ indicates that the starch with an ordered structure was more able to resist enzymes and acids. The short-range ordered degree affects the hydrolysis, digestion, and gelatinization of starch [64,65]. In the present study, starch from the control group presented the lowest R1045/1022 cm^−1^ value (0.66–0.71), followed by the nitrogen fertilizer treatment groups N1 (0.72) and N2 (0.73–0.78). The absorbance ratios of 1045/1022 cm^−1^ and 995/1022 cm^−1^ increased along with the nitrogen application levels (Table 1), indicating that nitrogen significantly promotes crystalline regions and boosts the packing or winding of helical structure formation. The short-range molecular order of double helix structures in starch is necessary for the formation of crystalline lamellae, while the long chains of amylopectin form a complete crystalline structure [11,60]. Starch in the N2 treatment showed a high absorbance ratio of 1045/1022 and 995/1022, suggesting that these starches exhibited more double-helix structures within the crystalline lamella of starch granules [63], which correspond with higher crystallinity inferred from XRD. However, crystallinity by XRD reflects the order of the whole granule structure, while the FTIR spectra showed structure characteristics only 2 μm deep from the granule surface [24,66]. 

#### 3.4.5. Amylopectin Chain-Length Distribution (CLD)

Amylopectin chains were grouped into A chains (DP 6–12), B1 chains (DP 12–24), B2 chains (DP 24–36), and B3 chains (DP > 36) [30,67]. The change in debranched amylopectin chain length distribution profiles with different nitrogen application levels is shown in Figure 6. Parameters of amylopectin were compiled and are depicted in Table 2. The study’s findings revealed that the distribution of amylopectin chain length peaked at DP12, regardless of the nitrogen fertilizer treatment. A difference in the chromatograms of all the treatments was not observed. 

Nitrogen fertilizer significantly decreased the average chain length, by increasing the proportion of short A and B1 chains while decreasing the B2 and B3 chain proportion slightly. This further elevated the (A + B1)/(B2 + B3) ratio, which was found at a high ratio in cereal crop starch (Bertoft, 2017). The differences between samples treated with N fertilizer and their controls N1–N0 and N2–N0 show the same trend, which is an increase in the B1 chain (DP 10–20) and a decrease in the proportion of mid to long chains. The change was observed more in N2–N0 than N1–N0. We can speculate that it might be due to the extension of the grain-filling period by the nitrogen treatment; most of the inferior grains developed late with poor plumpness, when the short chain branches had not developed into longer chain branched amylopectin. Short chains (A + B1 chains) form double helices that are involved in crystal formation in the starch granules [68] in accordance with the results for double-helix structure and crystalline structure above. Short A chains in amylopectin were positively correlated with relative crystallinity. The trend is more obvious in the N2 treatment compared with other treatments. The functional qualities of starch depend on the amylopectin fine structure, and the length of the amylopectin chain is closely related to the starch’s crystal structure and pasting properties [16,30,69]. 

#### 3.4.6. Pasting Properties of Starch by RVA

The RVA response curve and characteristic values correctly reflect the gelatinization properties of rice flour under hydrothermal conditions by visually depicting the molecular changes of starch granules during gelatinization and swelling [53]. The pasting properties of starch were significantly different for rice from the three treatments (Table 3). The peak viscosity, trough viscosity, breakdown, and final viscosity were significantly decreased with the increasing nitrogen rate, consistent with previous research [6,10]. The peak viscosity typically signifies the highest swelling degree of the starch when heating [70]. A higher peak viscosity shows a greater water absorption capacity and swelling ability of the starch, which may result in a more viscous texture of cooked rice. The reduced overall viscosity of starch with nitrogen treatment may not only be correlated with the decline in amylose content, but also ascribed to the proportion of B3 chains declining [71]. In Li’s report, the amylose content was positively related to the final viscosity [72]. Final viscosity refers to the ultimate viscosity achieved during gelatinization, which is closely associated with the stickiness and texture of rice. The results indicate that an increase in nitrogen fertilizer application leads to a reduction in the final viscosity of starch, suggesting that rice samples with nitrogen fertilization exhibit lower viscosity compared to the control group. This finding is consistent with the results obtained from TPA.

The gelatinization temperature (GT) is the minimum temperature for pasting starch, which is related to the stability of starch granules. The higher GT in the N2 treatment may be ascribed to the higher crystallinity structure, external short-range order molecular structure, and double helix structure of starch.

Eating and cooking quality was significantly positively correlated with breakdown and negatively correlated with setback viscosity values [53]. High breakdown viscosity is a negative parameter and shows the rupture of starch granules while heating and swelling in water during gelatinization [73,74]. Setback viscosity may be driven by the leached amylose rearrangement during cooling and reflects the retrogradation of the starch paste [75]. Inhibiting granule swelling and maintaining the swollen starch granules are both possible with amylose [76]. The high breakdown viscosity and low setback viscosity value indicated that cooked rice with the highest palatability was neither too hard nor too sticky, which is in accordance with the texture parameters in Section 3.3. The pasting properties in the N1 treatment resulted in better eating and cooking quality than that in N2.

#### 3.4.7. Thermal Properties of Starch by DSC

The thermal characteristics of starch in different nitrogen fertilizer treatment measured by DSC are illustrated in Table 3. The gelatinization temperatures (T0, Tp, Tc, and ΔT) and enthalpy were significantly increased along with the nitrogen fertilizer rates, thereby indicating the varying degree of crystallinity within the starch granules at different nitrogen fertilizer rates. Enthalpy reflects the energy required to gelatinize the starch. It is a measure of crystallinity and the loss of the double helix and molecular order in starch granules during paste heating [52]. The gelatinization temperature refers to the stability of the starch structure. With a large number of short chain amylopectin molecules, the starch in the N2 treatment may have formed more double helix structures, and, as a result, the starch granules were more compact and displayed a higher degree of molecular order. Although there was not a significant difference between the samples, it is possible that the more crystalline structures and higher degree of polymerization in amylopectin chains led to an increase in the melting point or gelatinization [77,78]. The gelatinization of starch has been shown to be influenced by interactive factors, including granule morphology, amylose/amylopectin ratio, the structure of starch molecules, and the content of minor components [79].

#### 3.4.8. Swelling Power and Solubility

Swelling power reflects the ability of starch granules to bind to water, and solubility signifies the dissolution of starch molecules in water at a certain temperature. These two parameters define the interaction between starch and water molecules and are responsible for several factors. The solubility of starch in the different treatments of the two varieties ranged from 16.5% to 21.7%, with a swelling range of 15–18.1%, both with an increasing trend with the application of nitrogen. Compared to the N0 treatment, the solubility and swelling of starch in the N2 treatment increased 1.58 and 2.93 percentage points on average, respectively. The highest levels of starch solubility were observed in the N2 treatments with lower amylose content. The granules’ swelling may have been caused by prior water uptake into the amorphous region, which contains a large proportion of amylose. Amylose’s double helix can inhibit the further access of solvents and swelling of granules and, therefore, hinder the hydration of amorphous regions and the entire starch granule [76,80,81]. In addition, both water solubility and swelling power are determined by the size and surface morphological characteristics of starch granules. A high swelling power was observed in starches with smaller granules because of their high affinity for water. The starch granules with nitrogen treatment showed uneven surfaces and holes or pores appeared on the granule surface, which made it easier for water molecules to permeate into the interior of the granules and combine with more starch molecules to form hydrogen bonds, resulting in an increase in swelling power and solubility (Figure 1 and Appendix A). Starch in the N2 treatment with more A-type granules had a larger surface area and more polar groups to combine with water, showing the highest solubility and swelling power. Whereas starches in the N0 treatment with a higher content of amylose and higher B3-chain content in the amylopectin showed a loose package of the double-helix structure and microcrystalline structure, leading to the suppression of the starch solubility and exhibiting a lower swelling power [29,82]. 

### 3.5. Relationship between Starch Structure Characteristics and Functional Properties 

The Pearson correlation coefficient for starch structure and properties is shown in a plot (Figure 7). As the proportion of amylopectin increased, more helical structures formed, which were the basis of the crystalline lamellae, and improved the degree of order and the relative crystallinity of starch granules [72]. That means the starch with high nitrogen treatment has a more stable structure. 

The gelatinization of starch has been found to be influenced by interactive factors, including granule morphology, amylose/amylopectin ratio, the microstructure of starch and the trace composition of the flour [79]. It has been shown [83] that the presence of short amylopectin chains (DP 6~12) induces lower gelatinization temperatures. Other literature has indicated that long chains formed stable double helices which span at least two crystalline lamellae, requiring a higher temperature to disorder the structure [84]. Tao [85] suggested that gelatinization temperatures decrease with DP < 60 amylopectin chains. In the present study, in contrast, the gelatinization temperatures rose with an increase in the proportion of amylopectin chain DP < 21. 

Starch granules swell during gelatinization and pasting. Amylose suppresses the granules’ expansion, and it has been considered as an acknowledged explanation for differences in eating quality. However, it is not that simple, and amylose content alone is insufficient to explain pasting properties. These may be attributed to super long debranched amylopectin, which, like amylose, leaches out and forms a network around the granules, inhibiting further swelling and maintaining the swollen starch granules [85]. The small granules with more surface area had a high affinity for water. The difference in the granule size and the fine structure of amylopectin affected the pasting and gelatinization properties, which may lead to the variations of texture and eating quality. When heated in hot water, proteins play an important role in gelatinization and the thermodynamic properties of starch, besides leached amylose which inhibited the starch granules from swelling. The increase in protein bodies in the high N-application treatment adhered to the surface of amyloplasts and inhibited starch gelatinization [86].

In fact, the structure, size, and fine distribution of starch granules were not separate from each other, but closely related to each other [22]. They contribute to the changes in starch water absorption, gelatinization, and thermal and other characteristics in synergy. The other components of grain also act in combination with starch during gelatinization. This is a rather complex yet integrated system, and, most of the time, multiple factors rather than one or two factors determine the physicochemical properties of starch.

## 4. Conclusions

Two cultivars of *japonica* rice were employed to investigate grain yield and eating quality and their associations with starch structure and properties at three nitrogen application rates. The nitrogen application level significantly affects the composition and structure of starch, regulating the starch properties and texture quality of cooked rice. With nitrogen treatments (N1 and N2) the yield was increased by 49~87%; starch granules’ size decreased, and the surface of the starch granules became uneven and porous. With a greater relative crystallinity and a higher external short-range order, the structure of long branch-chain amylopectin was lower, and the proportion of short branch amylopectin was higher. Gelatinization enthalpy and temperatures rose, and viscosity declined. All of these changes in structure and proportion led to a harder texture and deteriorated the eating and cooking quality. The change in functional properties contributed to multiple factors synergistically and antagonistically in terms of the structure and contents, such as the granule size, fine structure of amylopectin, compositions, and hierarchical structure. Overall, a moderate reduction in nitrogen application improves rice texture and starch quality without affecting grain yield.

## Figures and Tables

**Figure 1 foods-12-02601-f001:**
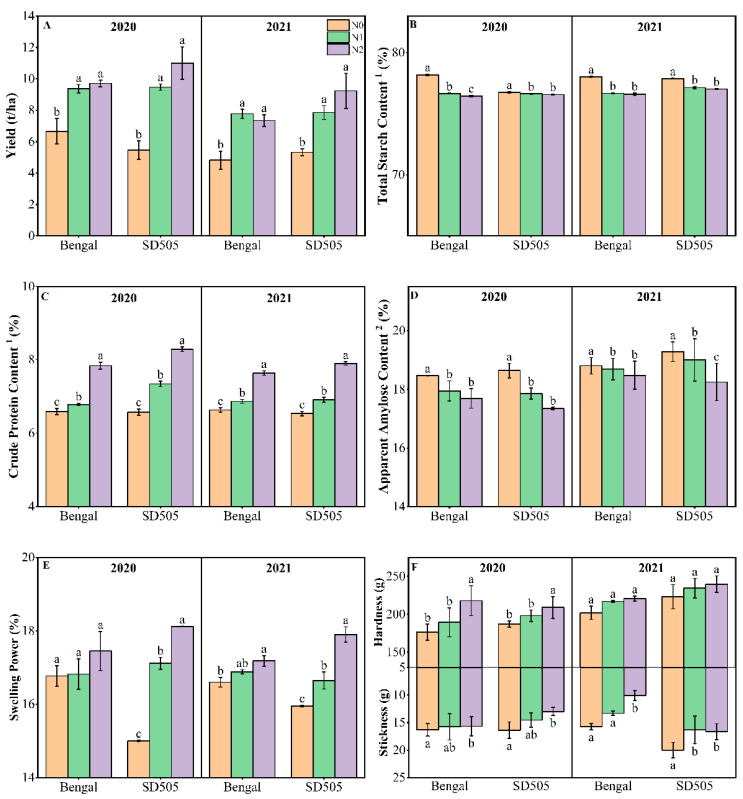
Effect of nitrogen application rate on yield (**A**), total starch ^1^ (**B**), protein ^2^ (**C**), amylose content of rice (**D**), swelling power (**E**), and TPA attributes (**F**). Mean ± SD values followed by the same letters do not differ significantly at *p* < 0.05. ^1^ Data expressed on the basis of white rice. ^2^ Data expressed on the basis of total starch content.

**Figure 2 foods-12-02601-f002:**
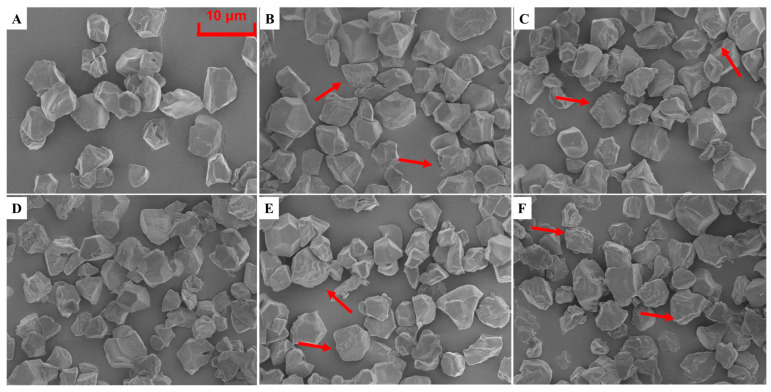
Scanning electric microscope (SEM) images of Bengal (**A**–**C**,**a**–**c**) and SD505 (**D**–**F**,**d**–**f**) starch in 2020 (**A**–**F**) and 2021 (**a**–**f**). N0 (**A**,**D**,**a**,**d**), N1 (**B**,**E**,**b**,**e**), and N2 (**C**,**F**,**c**,**f**). The arrows indicate the surface characteristics of starch granules.

**Figure 3 foods-12-02601-f003:**
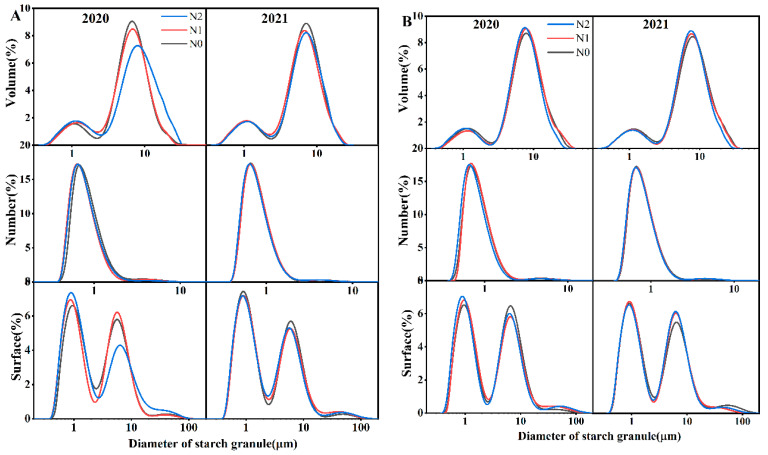
Effect of nitrogen rates on the granule size distribution of starch in volume, number, and surface percentage in Bengal (**A**) and SD 505 (**B**).

**Figure 4 foods-12-02601-f004:**
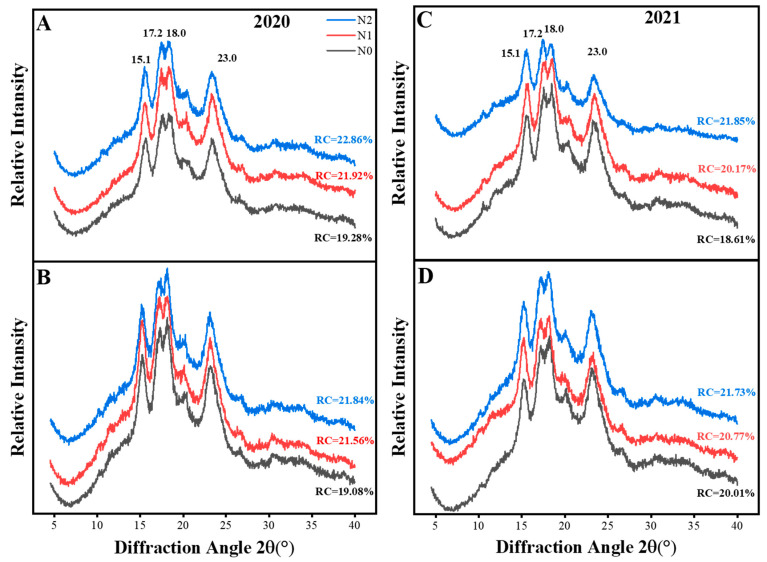
X-ray diffraction patterns of starches at different nitrogen rates from Bengal (**A**,**C**) and SD505 (**B**,**D**).

**Figure 5 foods-12-02601-f005:**
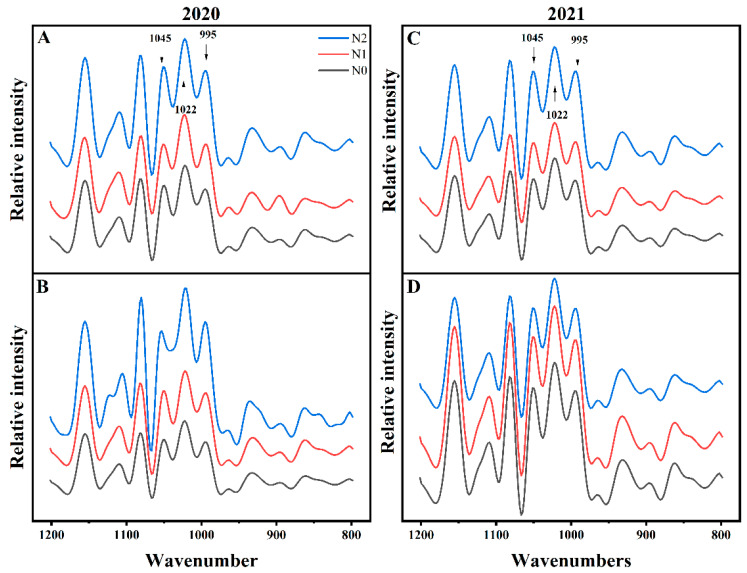
Deconvolved FTIR profile at different nitrogen rates from Bengal (**A**,**C**) and SD505 (**B**,**D**) in 2020 (**A**,**B**) and 2021 (**C**,**D**).

**Figure 6 foods-12-02601-f006:**
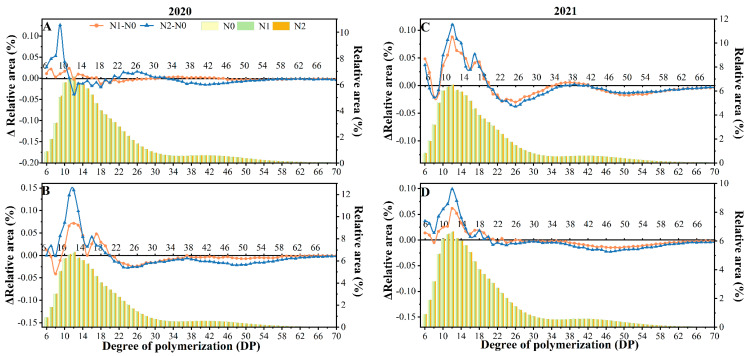
Changes in the chain-length distribution of amylopectin from Bengal (**A**,**C**) and SD505 (**B**,**D**) in response to N application levels. N1–N0, N2–N0 represent the differences between samples treated with N fertilizer (N1 and N2) and their control (N0), respectively.

**Figure 7 foods-12-02601-f007:**
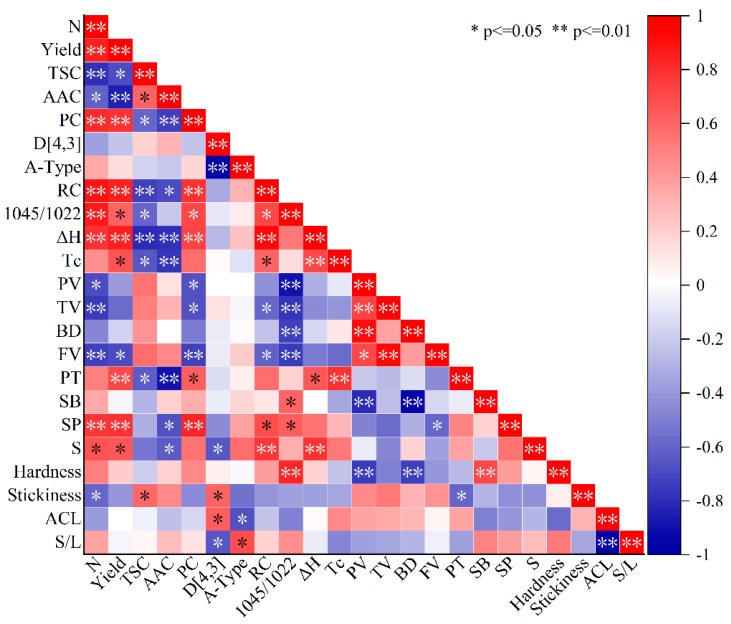
Correlations analysis of the structural and physicochemical properties and grain composition of *japonica* starches. N, nitrogen application rate; TSC, total starch content; AAC, apparent amylose content; PC, protein content; D [4, 3], volume-weighted mean diameter; A-type, small granule starch proportions; RC, relative crystallinity; 1045/1022, ratio of 1045/1022 cm^−1^; ΔH, gelatinization enthalpy; Tc, concluding gelatinization temperature; PV, peak viscosity; TV, trough viscosity; BD, breakdown viscosity; FV, final viscosity; PT, pasting temperature; SB, setback viscosity; SP, swelling power; S, water solubility; ACL, average chain length; S/L, the ratio of short chains and long chains.

**Table 1 foods-12-02601-t001:** Granule size distribution and FTIR ratio of starch at different nitrogen rates.

Year	Cultivars	Treatment	D [3, 2] (μm)	D [4, 3] (μm)	d (0.1) (μm)	d (0.5) (μm)	d (0.9) (μm)	A-Type (%)	B-Type (%)	1045/1022 (cm^−1^)	995/1022 (cm^−1^)
2020	Bengal	N0	4.26 ± 0.12 c	7.33 ± 0.15 c	1.2 ± 0.04 e	6.5 ± 0.01 d	11.76 ± 0.04 c	14.75 ± 0.05 c	85.25 ± 0.05 d	0.66 ± 0.02 c	0.65 ± 0.01 d
		N1	4.04 ± 0.01 d	6.85 ± 0.08 d	1.29 ± 0.03 d	5.83 ± 0.05 e	11.87 ± 0.06 c	16.62 ± 0.01 b	83.38 ± 0.01 e	0.72 ± 0 b	0.67 ± 0 c
		N2	4.85 ± 0.07 b	6.76 ± 0.14 d	1.27 ± 0 d	5.74 ± 0.02 f	12.38 ± 0.29 b	17.81 ± 0.02 a	82.19 ± 0.02 f	0.73 ± 0.01 b	0.7 ± 0 b
	SD505	N0	4.92 ± 0 b	8.54 ± 0.01 a	1.74 ± 0 a	7.45 ± 0.02 a	13.62 ± 0.14 a	11.28 ± 0.07 f	88.72 ± 0.07 a	0.68 ± 0.02 c	0.71 ± 0.01 b
		N1	5.5 ± 0.05 a	8.39 ± 0.01 a	1.51 ± 0.01 b	7.12 ± 0 b	13.33 ± 0.18 a	12.57 ± 0.07 e	87.43 ± 0.07 b	0.73 ± 0.01 b	0.74 ± 0.01 a
		N2	5.55 ± 0.01 a	7.63 ± 0.07 b	1.37 ± 0.02 d	6.85 ± 0.04 c	13.49 ± 0.31 a	13.51 ± 0.05 d	86.49 ± 0.05 c	0.76 ± 0 a	0.76 ± 0 a
2021	Bengal	N0	4.31 ± 0.03 f	7.21 ± 0.05 c	1.26 ± 0.01 c	6.13 ± 0.03 d	11.82 ± 0.01 c	16.08 ± 0.03 c	83.92 ± 0.03 d	0.68 ± 0.02 d	0.65 ± 0.01 d
		N1	4.82 ± 0.03 d	6.95 ± 0.16 d	1.24 ± 0.04 cd	5.81 ± 0.02 e	11.94 ± 0.03 c	16.83 ± 0.02 b	83.17 ± 0.02 e	0.74 ± 0.01 b	0.72 ± 0.01 c
		N2	4.69 ± 0.02 e	6.67 ± 0.19 e	1.2 ± 0.02 d	5.77 ± 0.02 e	12.45 ± 0.03 b	17.73 ± 0.05 a	82.27 ± 0.05 f	0.77 ± 0 ab	0.74 ± 0 b
	SD505	N0	6.28 ± 0.04 a	8.45 ± 0.04 a	1.53 ± 0.01 a	6.93 ± 0.02 a	12.72 ± 0.01 b	12.63 ± 0.04 f	87.37 ± 0.04 a	0.71 ± 0.03 c	0.71 ± 0.01 c
		N1	4.9 ± 0.04 c	8.21 ± 0.02 a	1.49 ± 0.02 ab	6.72 ± 0.04 b	13.42 ± 0.42 a	12.77 ± 0.01 e	87.23 ± 0.01 b	0.76 ± 0.01 ab	0.74 ± 0 b
		N2	5.2 ± 0.02 b	7.78 ± 0.06 b	1.46 ± 0.01 b	6.38 ± 0.08 c	13.74 ± 0.07 a	13.33 ± 0.05 d	86.67 ± 0.05 c	0.78 ± 0.01 a	0.75 ± 0 a

The cumulative diameter values of d (0.1), d (0.5), and d (0.9) signify the diameter at which 10, 50, and 90% of particles, respectively, fell within the specific size range. D [3, 2] and D [4, 3] were calculated by mean diameter based on surface and volume, respectively. Values are present as means ± SD (*n* = 3); different letters within the same column differ significantly (*p* < 0.05).

**Table 2 foods-12-02601-t002:** Distribution of amylopectin at different nitrogen application levels of rice.

Cultivars	Year	Treatment	A (%)	B1 (%)	B2 (%)	B3 (%)	ACL (DP)	A + B1/B2 + B3	A + B1 (%)
Bengal	2020	N0	29.93 e	49.22 c	10.70 b	10.14 a	19.44 a	3.80 e	79.15 f
		N1	30.01 d	49.19 d	10.71 b	10.09 b	19.40 b	3.81 e	79.20 d
		N2	30.18 c	49.15 e	10.75 a	9.91 c	19.34 c	3.84 d	79.34 d
	2021	N0	30.19 c	49.32 f	10.54 c	9.95 c	19.31 d	3.88 c	79.51 c
		N1	30.31 b	49.49 b	10.38 d	9.83 d	19.23 e	3.95 b	79.80 b
		N2	30.61 a	49.49 a	10.34 e	9.55 e	19.12 f	4.03 a	80.10 a
SD505	2020	N0	29.59 f	48.64 d	10.93 a	10.84 a	19.72 a	3.59 f	78.23 f
		N1	29.80 e	48.83 c	10.84 ab	10.52 b	19.56 b	3.68 e	78.63 e
		N2	29.83 d	48.90 b	10.75 b	10.51 b	19.57 b	3.70 d	78.73 d
	2021	N0	30.16 c	48.95 b	10.60 c	10.28 c	19.43 c	3.79 c	79.11 c
		N1	30.32 b	49.11 a	10.58 c	9.98 d	19.32 d	3.86 b	79.44 b
		N2	30.54 a	49.11 a	10.54 c	9.81 e	19.23 e	3.91 a	79.64 a

Values are presented as the means of three repetitions; different letters within the same column and same year differ significantly (*p* < 0.05).

**Table 3 foods-12-02601-t003:** Effects of nitrogen fertilizer on the pasting properties and thermal properties of rice starch in 2020 and 2021.

Year	Cultivars	Treatment	Pasting Properties						Thermal Properties			
			PV (cp)	TV (cp)	BD (cp)	FV (cp)	SB (cp)	PT (℃)	ΔH (J/g)	T0 (℃)	Tp (℃)	Tc (℃)
2020	Bengal	N0	3303 ± 22.23 a	1874 ± 37.67 a	1429 ± 15.51 a	3001 ± 46.68 a	1127 ± 12.33 a	71.27 ± 0.45 b	6.88 ± 0.08 e	61.28 ± 1.02 c	68.70 ± 0.22 bc	76.23 ± 0.09 d
		N1	3165.67 ± 78.41 ab	1883.33 ± 101.21 a	1282.33 ± 120.48 ab	2996 ± 83.33 a	1112.67 ± 22.4 a	71.88 ± 0.02 ab	7.51 ± 0.07 b	62.13 ± 0.87 abc	68.93 ± 0.17 ab	76.37 ± 0.12 d
		N2	3028.33 ± 124.63 bc	1876.67 ± 87.75 a	1151.67 ± 44.31 bc	3013.67 ± 137.91 a	1137 ± 50.46 a	71.88 ± 0.02 ab	7.67 ± 0.08 a	62.53 ± 0.54 abc	69.17 ± 0.12 ab	76.90 ± 0.22 b
	SD505	N0	3040.33 ± 44.73 bc	1899.33 ± 67.37 a	1141 ± 84.75 bc	2994.67 ± 50.61 a	1095.33 ± 105 a	71.50 ± 0.39 b	7.12 ± 0.07 d	61.56 ± 0.49 bc	68.10 ± 0.08 d	76.63 ± 0.12 c
		N1	2955.33 ± 67.43 c	1859 ± 24.71 a	1096.33 ± 69.43 cd	2924.67 ± 17.44 ab	1065.67 ± 28.41 a	72.63 ± 0.67 a	7.31 ± 0.02 c	63.03 ± 0.41 ab	68.57 ± 0.17 c	77.00 ± 0.08 b
		N2	2763.33 ± 47.13 d	1791 ± 5.35 a	972.33 ± 42.28 d	2791 ± 89.29 b	1000 ± 89.5 a	72.60 ± 0.04 a	7.48 ± 0.06 b	63.63 ± 0.25 a	68.97 ± 0.17 ab	77.17 ± 0.12 a
2021	Bengal	N0	3063.33 ± 20.4 a	1985 ± 32.95 a	1078.33 ± 51.19 ab	3169.67 ± 25.95 a	1184.67 ± 17.25 a	71.22 ± 0.34 ab	6.83 ± 0.05 e	62.47 ± 0.05 a	68.17 ± 0.09 b	74.43 ± 0.26 d
		N1	2836.67 ± 34.74 bc	1863.33 ± 24.85 bc	973.33 ± 39.19 c	2983.67 ± 38.85 ab	1120.33 ± 21.3 bc	71.23 ± 0.4 ab	7.07 ± 0.05 cd	62.33 ± 0.17 a	68.47 ± 0.25 b	75.33 ± 0.17 c
		N2	2721 ± 20.4 d	1730.33 ± 13.89 d	990.67 ± 17.15 c	2863.33 ± 33.08 d	1133 ± 21.23 b	71.78 ± 0.02 a	7.27 ± 0.12 ab	62.70 ± 0.22 a	69.03 ± 0.12 a	76.43 ± 0.21 a
	SD505	N0	3063.33 ± 91.58 a	1961 ± 111.08 ab	1102.33 ± 35.37 a	3069 ± 104.01 ab	1108 ± 10.2 bc	70.70 ± 0.39 b	6.96 ± 0.11 de	62.37 ± 0.17 a	69.10 ± 0.08 a	75.37 ± 0.12 c
		N1	2856.33 ± 28.22 b	1773.33 ± 52.56 cd	1083 ± 33.08 ab	2892 ± 41.98 bc	1118.67 ± 14.38 bc	70.95 ± 0.04 b	7.14 ± 0.02 bc	62.37 ± 0.12 a	69.30 ± 0.08 a	75.63 ± 0.21 bc
		N2	2752.33 ± 20.37 cd	1742.33 ± 8.06 cd	1010 ± 24.91 bc	2821 ± 22.2 d	1078.67 ± 25.93 c	71.28 ± 0.33 ab	7.36 ± 0.06 a	62.77 ± 0.05 a	69.40 ± 0.24 a	75.87 ± 0.05 bc

Values are presented as means ± SD (n = 3); different letters within the same column and same year differ significantly (*p* < 0.05).

## Data Availability

The data used to support the findings of this study can be made available by the corresponding author upon request.

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
