# Peer review of "Moderate Reduction in Nitrogen Fertilizer Results in Improved Rice Quality by Affecting Starch Properties without Causing Yield Loss"

_foods, 2023, doi:10.3390/foods12132601_

Round 1

Reviewer 1 Report

The article makes an important contribution to the scientific community, considering that it deals with the application of nitrogen in two rice varieties. The results are well discussed and show that a moderate reduction in nitrogen application improves rice texture and starch quality without affecting grain yield.

To improve the work, here are some corrections:

1) Unit corrections were made to standardize the entire work (attached)

2) In the methodology, if possible, try to reduce the text 2.2 and 2.4 so that it is similar to the other items

3) The conclusions should be more objective since throughout the work the results were discussed in an adequate and coherent way.

Reviewer 2 Report

The manuscript “Moderate reduction in nitrogen fertilizer results in improved rice quality by affecting starch properties without causing yield” presents an interesting and well designed work, with consistent and well-presented results. The conclusions are very relevant.

Some minor remarks:

- Ln 218-219: This sentence is not clear, please rephrase.

- Fig 1: Please, check significance letters, they seem not coherent.

- Ln 245-274: in figure 1, total starch is calculated relative to flour, but ACC is calculated relative to starch, so why do authors relate the decrease in ACC to the decrease in total starch?

- Ln 252: “as shown in section 3.4.6 and 3.4.8. AAC” I think this can be deleted.

- Ln 259-261. This information is repeated.

- Ln 273-275: Please, check the wording.

- section 3.4.6. Neither tasting properties nor cooking properties were included in Materials and methods.

- Ln 474: may authors mean “lower amylose content”?

- Ln 499: Please, check the wording “starch molecules and minor-component of contents”.

- Ln 549: Please, check the wording “fine structure of starch, content of component”.

Reviewer 3 Report

The manuscript describes the influence of nitrogen fertilizers on rice grain and starch quality. The topic is interesting and the manuscript is well-structured. However, it is not acceptable in its present form and should be revised:

Comments: 

- line 134: Usually rice flour is used for texture measurement; why you have used rice grains? Why you have used texture profile analysis (TPA) ? The cooked grains have a granulated and non-cohesive structure and TPA is not suitable for rice grains. Compression is a better test for rice grains. Could you please add a reference for this method?  

line 191: Why the maximum heating temperature was 95°C in DSC? Usually the maximum temperatures ≥120°C are applied in this experiment. Please add a reference for this method.

- Figure 2: The magnification of images is very low and it is difficult to distinguish between the images. Please use larger images.

Line 337: relative crystallinity (Please apply in other parts of the article)

Table 2: It seems that the statistical analysis is not correct. Please recheck the results.

Under the Table 2: Please write that the values are the average of how many replications and also explain the meaning of different letters.

Line 430: Please explain what does "peak viscosity" show. You can use the following manuscript: 10.3390/gels8110693

Lines 434-435: High breakdown viscosity It is a negative parameter and shows the rupture of starch granules not the ability to resist rupture. Please correct these sentences.

line 445: Please explain the "final viscosity" and the influence of fertilizers on this parameter. 

Table 3: Please recheck the setback values.

The language is fine.

Round 2

Reviewer 3 Report

The manuscript is acceptable.

Author Response

We appreciate very much for your recognition of our research work. Thank you for your precious time on this manuscript.